# The Effect of Maternal Probiotic or Synbiotic Supplementation on Sow and Offspring Gastrointestinal Microbiota, Health, and Performance

**DOI:** 10.3390/ani13192996

**Published:** 2023-09-22

**Authors:** Dillon P. Kiernan, John V. O’Doherty, Torres Sweeney

**Affiliations:** 1School of Veterinary Medicine, University College Dublin, D04 C1P1 Dublin, Ireland; dillon.kiernan@ucdconnect.ie; 2School of Agriculture and Food Science, University College Dublin, D04 C1P1 Dublin, Ireland; john.vodoherty@ucd.ie

**Keywords:** maternal transmission, microbiota, gut health, intestinal dysfunction, pathogen infection, beneficial bacteria, microbiota metabolites, swine

## Abstract

**Simple Summary:**

The gastrointestinal tract (GIT) microbiota refers to the collection of microorganisms that colonize the GIT. The composition and diversity of the GIT microbiota play a fundamental role in animal health and performance and is an area that is becoming increasingly important with recent introduction of new restrictions on antibiotic and antimicrobial use in animal production. Enhancing the establishment of the GIT microbiota in the initial stages of life can improve health and performance, both in the immediate period and later life. Interestingly, the offspring’s microbiota is heavily influenced by the maternal microbiota, thereby underlining modulation of the maternal microbiota as a promising strategy to enhance the health and growth of the offspring. A probiotic is a beneficial live microorganism that, when supplemented, has beneficial effects on the host’s health. A synbiotic is a combination of a probiotic and another bioactive known as a prebiotic, which is essentially a substrate for beneficial microbes in the gut. Supplementing probiotics or synbiotics in the maternal diet presents a promising strategy to firstly modulate the maternal GIT microbiota, conferring several health benefits to the sow, and secondly to influence the establishment of the offspring’s GIT microbiota, conferring several health benefits to the offspring. The current review discusses the importance and suggested mechanisms of action for maternal probiotic and synbiotic supplementation and details key findings from studies that have investigated the effects of maternal probiotic or synbiotic supplementation in pigs.

**Abstract:**

The increasing prevalence of antimicrobial-resistant pathogens has prompted the reduction in antibiotic and antimicrobial use in commercial pig production. This has led to increased research efforts to identify alternative dietary interventions to support the health and development of the pig. The crucial role of the GIT microbiota in animal health and performance is becoming increasingly evident. Hence, promoting an improved GIT microbiota, particularly the pioneer microbiota in the young pig, is a fundamental focus. Recent research has indicated that the sow’s GIT microbiota is a significant contributor to the development of the offspring’s microbiota. Thus, dietary manipulation of the sow’s microbiota with probiotics or synbiotics, before farrowing and during lactation, is a compelling area of exploration. This review aims to identify the potential health benefits of maternal probiotic or synbiotic supplementation to both the sow and her offspring and to explore their possible modes of action. Finally, the results of maternal sow probiotic and synbiotic supplementation studies are collated and summarized. Maternal probiotic or synbiotic supplementation offers an effective strategy to modulate the sow’s microbiota and thereby enhance the formation of a health-promoting pioneer microbiota in the offspring. In addition, this strategy can potentially reduce oxidative stress and inflammation in the sow and her offspring, enhance the immune potential of the milk, the immune system development in the offspring, and the sow’s feed intake during lactation. Although many studies have used probiotics in the maternal sow diet, the most effective probiotic or probiotic blends remain unclear. To this extent, further direct comparative investigations using different probiotics are warranted to advance the current understanding in this area. Moreover, the number of investigations supplementing synbiotics in the maternal sow diet is limited and is an area where further exploration is warranted.

## 1. Introduction

The increasing concern around the development of antimicrobial resistant pathogens has prompted a goal of reducing antibiotic and antimicrobial use in commercial pig production. Most recently, the EU has begun to phase out the use of pharmacological levels of Zinc Oxide (ZnO) (Commission Implementing Decision of 26.6.2017, C (2017) 4529 Final), while additional restrictions on the use of infeed medication have been applied (Regulation (EU) 2019/6). Canada is currently reviewing the implementation of similar restrictions on the use of ZnO (Canadian Food Inspection Agency, Feed Regulations 2022). China have also implemented restrictions on the use of ZnO, although they still allow a relatively high inclusion rate for the initial two weeks postweaning. Zinc oxide is an extremely effective antimicrobial that has played an integral part in alleviating the gut related pathologies and dysbiosis associated with weaning on commercial pig farms. These new restrictions are driving research into alternative dietary interventions which support the health and development of the pig. To date, the majority of research has focused on postweaning dietary manipulation; however, maternal dietary intervention provides a promising alternative. Supplementing the sow’s diet during gestation and lactation can enhance gut development and health in the offspring prior to weaning, thereby contributing to improved health and performance, in the immediate postweaning phase and beyond [1,2,3,4,5].

To meet the economic demand on commercial farms, weaning occurs at approximately three to four weeks of age. The weaning period is stressful, as the piglet must cope with changes in diet, environment, and social conditions. This combination of factors places stress on the piglet, which can disrupt the GIT microbial ecosystem [6]. Additionally, the change from a liquid-based milk diet to a solid-based plant diet switches the dietary substrates available to microbes in the GIT, with inevitable disruption to the microbial ecosystem [7]. This transition changes the microbiota composition as it evolves from a milk-oriented microbiota to a plant-oriented microbiota and presents an opportunity for enteric pathogens to colonize the GIT as it deals with the shift in feed type. The weaning transition period is also characterized by a reduction in feed intake in the initial days postweaning, further confounding other stressors associated with weaning [8,9]. The potential of dietary interventions as a means of promoting a healthy microbiota composition during the postweaning period aligns to current research and is also in line with the recent legislative constraints. The limited feed intake in the immediate days postweaning suggests that a better strategy to manage the immediate postweaning phase would be the establishment of a healthy microbiota in the suckling pig prior to weaning. The relative abundance of certain groups in the suckling pig’s microbiota, such as *Lactobacillaceae*, *Ruminococcaceae*, *Lachnospiraceae*, and *Prevotellaceae*, are associated with a reduction in the occurrence of diarrhoea in the pig postweaning [10]. Given that the sow is the predominant contributor to the shaping of the offspring’s microbiota, dietary manipulation to modulate the sow’s microbiota, prior to farrowing and during lactation, is a compelling area of exploration. Maternal probiotic supplementation can improve the lifetime performance of the offspring, further highlighting the suckling period and the sow’s microbiota as critical research target areas [2].

Microbial communities differ in composition and density along the GIT, as well as across the mucosal and luminal region [11]. Many studies to date focus on the microbiota of the faeces as it is easy to obtain and does not require the animal to be euthanized. However, specific alterations in the microbiota along the GIT are not captured when solely analysing faeces [12]. In studies where animals are euthanized, it is common to collect samples from the ileum and colon and in some cases the jejunum [13,14,15]. The analysis of the microbiota composition of the ileum, and to a lesser extent the jejunum, is of particular importance in the postweaned pig as the pathogenesis of postweaning diarrhoea is associated with the colonization of the small intestine by Enterotoxigenic *Escherichia coli* (ETEC) and other diarrhoeagenic *Escherichia coli* strains (as reviewed in [16]). In research studies, modulation of the GIT microbiota is typically assessed by analysing the microbial composition of the GIT microbiome, quantifying the abundance of bacterial groups, and evaluating microbial diversity [17,18,19,20]. The concentration of beneficial bacterial metabolites is generally analysed as indicators of the level and type of fermentation occurring in the GIT [17,18,19,20,21]. Furthermore, in terms of assessing gut health, particular attention is paid to the intestinal epithelial barrier integrity and the villous structures and crypt depths in the small intestine [3,18,19,22]. 

The diversity of composition of the GIT microbiota is influenced by a number of factors such as diet, age, and body weight [23,24]. The GIT microbiota plays a key role in feed efficiency [25], growth performance [23], and disease [26] in pigs. Targeting improvements in the GIT microbiota via dietary intervention has proven effective in pigs at all stages of development, improving average daily gain, food conversion ratio (FCR), and overall animal health [27,28,29]. Similarly, in broilers, microbiota modulation can promote improved average daily gain, FCR and overall animal health [30] and in laying hen diets it can be associated with enhanced egg production parameters and FCR [31]. This demonstrates the role of the GIT microbiota and the potential for dietary intervention to improve performance and efficiency of monogastric animals. Bioactives that have microbiota modulating capabilities can be classified by their mode of action (Figure 1). The classifications include, but are not limited to, probiotics, prebiotics, synbiotics, stimbiotics, postbiotics, and certain plant derivatives. This review discusses the importance and suggested mechanisms of action for maternal probiotic and synbiotic supplementation and details key findings from studies that have investigated the effects of maternal probiotic or synbiotic supplementation in pigs.

## 2. Probiotics

A probiotic is defined as a “live microorganism which when administered in adequate amounts confers a health benefit on the host” [32]. Microbes best adapted to occupy a particular niche within the intestinal tract will eventually dominate it. An established and mature healthy microbiota is one in which all environmental niches are filled with beneficial or nonpathogenic microbes. This makes it difficult for opportunistic pathogens to colonize the GIT and the host more resistant to infection by pathogens. This phenomenon is referred to as ‘bacterial antagonism’. Aside from the suppression of pathogenic proliferation, probiotic bacteria, through fermentation, produce health-promoting compounds including short-chain fatty acids (SCFA), such as acetate, propionate, butyrate, and organic acids, such as lactate, succinate, and pyruvate [33,34,35,36]. Furthermore, many metabolites produced by probiotic bacteria have beneficial immune modulating capabilities [37,38,39]. A probiotic’s general mode of action is to introduce beneficial bacterial strains into the GIT environment which then fill environmental niches. This in turn leads to an increase the production of host-health promoting metabolites and enhanced microbiota diversity and host resistance to pathogens, thus improving overall host health [40,41] (See Figure 1). 

A probiotic must be able to tolerate exposure to the gastric acid and bile salts in the proximal intestine in order to colonize the distal intestine. It must also have the capacity to limit the growth of pathogenic bacteria. Probiotics are selected and produced in several different forms based on the above criteria. Probiotics can be (adapted from [42]):(a)Single or multistrain [43];(b)Bacterial or nonbacterial: the predominant probiotics are bacterial; however, certain strains of yeast can be included [44];(c)Spore-forming or nonspore forming: in recent years, spore-forming bacteria have gained increased attention over nonspore-forming bacteria due to their stability and resilience, meaning they are more suitable for storage and more stable during animal feed handling procedures [45];(d)Allochthonous or autochthonous probiotics: allochthonous probiotics are microbes that are not usually in the gastrointestinal tract of the animal (e.g., Yeast and *Bacillus* species), while autochthonous probiotics are commensal bacteria that can be found in the animals gastrointestinal tract (e.g., *Bifidobacterium animalis* and *Lactobacillus plantarum*) [42].

The most commonly used microorganisms in probiotics are lactic acid-producing bacteria belonging to the *Lactobacillus* [43] and *Bifibacterium* [46] species, bacillus strains [47] and yeast *Saccharomyces* cerevisiae [44]. *Enterococcus* bacteria, more precisely *Enterococcus faecium* and *Enterococcus faecalis*, are also commonly used as probiotics [48]. Lactic acid-producing bacteria, or LAB, complexes are often produced as probiotic blends and can comprise various genera of LAB such as *Lactobacillus*, *Bifidobacterium*, *Lactococcus*, *Enterococcus*, *Pediococcus,* and *Streptococcus* [42]. *Bacillus* strains are not referred to as LAB, but several *Bacillus* species have a strong capacity to produce lactic acid [49,50,51]. To date, there have been several investigations utilizing a wide range of different probiotic species and strains to positive affect, both on their own and in probiotic blends. Given the modes of action of probiotics, certain strains may be more effective at particular stages of development. However, there remains some ambiguity in this regard and therefore is an area that requires increased research focus in the coming years. The characteristics of common probiotics that have been studied in the pig are presented in Table 1.

### 2.1. Beneficial Effects of Probiotic Supplementation

There are numerous mechanisms through which probiotics can confer health benefits to the host, and these can be both direct, via the probiotic itself, and/or indirect, via the production of beneficial metabolites and secretions, referred to as postbiotics (Figure 1). The following is a description of the beneficial modes of action depicted in Figure 1.

#### 2.1.1. Competitive Exclusion of Pathogenic Bacteria

The initial stages of intestinal infection involve the pathogen adhering to the intestinal mucosal surfaces. Certain probiotic bacteria compete with pathogenic bacteria for these crucial adhesion sites. Probiotics disrupt pathogen adhesion using several mechanisms, including: the formation of an auto-aggregation barrier, which occurs when bacterial cells adhere to each other [69,70]; competitive displacement, which occurs when probiotic bacteria outcompete pathogens for adhesion sites [69,71]; and co-aggregation, which occurs when probiotic bacteria attach with pathogens via specific molecules and prevent intestinal surface colonization [70,72,73]. Competitive displacement occurs when the probiotic outcompetes the pathogen by hindering its attachment, competing for nutrients, and reducing the fitness of competing bacteria via its antimicrobial factors [71]. The ability of a probiotic to adhere to the mucosa varies depending on the strain, its location in the intestine, and existing competition for adhesion sites [70,74,75]. The majority of strains exhibit significant pathogenic adhesion inhibitory effects; however, certain strains, such as *Lactobacillus rhamnosus*, are more effective than others [41,74,76]. 

As mentioned, one method for competitive displacement of pathogenic bacteria by probiotic bacteria involves competition for nutrients in the GIT. The regular supply of nutrients into the GIT means there are generally sufficient quantities to support the growth of most bacteria; however, for competitive exclusion to occur, there need only be competition for a single essential nutrient. For example, *Salmonella Typhimurium* is a pathogen that thrives in an inflamed GIT. It has adapted to acquire essential micronutrient metals in limited availability [77]. Administering the probiotic *Escherichia coli Nissle*, which assimilates iron in a similar mechanism to *Salmonella Typhimurium*, limits the ability of the pathogen to colonize the GIT [63,78]. The probiotics *Lactobacillus amylovorus* and *Lactobacillus mucosae* do not suppress the growth of salmonella at any location in the GIT, as they cannot outcompete *Salmonella Typhimurium* for iron [63]. These studies highlight the process of competitive exclusion via competition for a single crucial nutrient. 

#### 2.1.2. Antimicrobial Production

Another desirable property of a probiotic strain is its ability to limit the growth of pathogenic bacteria via metabolite secretions. The supernatant of the probiotic can contain a mixture of metabolite secretions, such as organic acids and bacteriocins, that inhibit the growth of pathogens. The production of organic acids reduces the pH of the surrounding environment which favours the growth of beneficial microbes, while inhibiting a range of harmful microbes [79,80]. Bacteriocins are specialized antimicrobial proteins produced by certain strains to inhibit pathogen growth. *Lactobacillus* and *Bacillus* species for example, produce a range of antimicrobial substances, such as bacteriocins and antifungal metabolites, that reduce the growth of pathogenic bacteria [41,81,82,83,84]. 

In addition to reducing the pH and producing antimicrobial molecules, certain probiotics can modulate the expression of virulence genes in pathogenic bacteria. Quorum sensing is a process of cell-to-cell communication that allows bacteria to regulate their gene expression in response to the cell population density [85,86]. In response to high cell densities, bacteria produce signalling molecules known as autoinducers. Pathogenic bacteria control the expression of their virulence genes by producing these autoinducers. Certain probiotics can secrete enzymes, known as quorum quenching enzymes, that degrade these autoinducers that signal increases in the expression of virulence genes in pathogenic bacteria [87,88,89,90]. Through quorum quenching, certain probiotic bacteria can control the proliferation of pathogenic bacteria and significantly modulate the GIT microbiota [90].

#### 2.1.3. Inhibiting Pathogenic Toxins

Toxins secreted by pathogenic bacteria, such as Shiga toxins and enterotoxins, can directly lead to intestinal dysfunction, including loss of water and electrolytes. These toxins can negatively affect barrier function by a number of different mechanisms, including: manipulation of the tight junction and cytoskeletal proteins [91,92]; passing through the enterocytes (via endocytosis, internalization, or the oligomerization of toxins) to form pores in the intestinal cell membranes [93]; and altering the isotonic stability in the cells of the intestine [94]. Certain probiotics can neutralize or inhibit toxins produced by pathogenic bacteria. *Lactobacillus zeae* inhibits ETEC enterotoxin production [95], while *Lactobacillus kefir* can reduce the cytopathic effect of *Clostridium difficile* [96]. Furthermore, *Lactobacillus*, *Pediococcus*, and *Bifidobacterium* strains downregulate the expression of the *Escherichia coli* toxin, Shiga toxin 2 (*STX2A*) gene, driven by the production of organic acids by the probiotic strains and subsequent decrease in pH [97].

#### 2.1.4. Short-Chain Fatty Acid Production

Short-chain fatty acids, mainly acetate, propionate, and butyrate, are organic acids that are predominantly produced in the GIT through anaerobic bacterial fermentation of undigested dietary fibre. The SCFA have numerous beneficial effects, including, promoting GIT barrier function, providing an energy source for epithelial cells, maintaining immune homeostasis, and modulating the GIT microbiota [98,99]. The quantity and type of SCFA produced depends on the physical and chemical properties of the substrate available to the probiotic coupled with the existing microbial composition of the GIT [100]. Increasing concentrations of SCFA reduces the pH, creating an unfavourable growth environment for pathogenic bacteria [79,80]. Butyrate is also a primary energy source for epithelial cells and has a crucial role in supporting colonic cell proliferation and intestinal growth [101,102]. The SCFA support multiple aspects of the GIT immune system, including: improved intestinal barrier function [37], including increased expression of β-defensins and cathelicidins [103], and immune responses, such as maintaining T cell homeostasis [39] and accelerating the differentiation of naïve T cells to Treg cells [104]. The regulation of the immune cells by SCFA is reviewed in detail by Liu et al. [105]. This immune modification by SFCA is suggested to be mediated via cell surface G protein-coupled receptors and inhibition of histone deacetylase and potential activation of histone acyltransferase [39,106,107]. The modification of epigenetic landmarks, such as histone deacetylase and acyltransferase, modulate the host’s gene expression. The current understanding and knowledge of epigenetic effects of SCFA on cells of the immune has been recently reviewed [108,109].

#### 2.1.5. Microbiota Derived Aryl Hydrocarbon Receptor Ligands

Aryl hydrocarbon receptor ligands are another important and beneficial postbiotic metabolite group [110]. The AhR is expressed in both immune and nonimmune cells of the GIT and has emerged as a key player in the regulation of intestinal homeostasis (as reviewed in [111]). There are numerous sources of endogenous and exogenous AhR ligands, one endogenous source being the GIT microbiota itself. These AhR ligands are typically produced by the microbiota via metabolism of tryptophan [110]. The microbiota produces a variety of AhR ligands, which are typically indole derivatives such as indole-3 ethanol (IE), indole-3 pyruvate [112], indole-3 aldehyde (I3A), and tryptamine (TA) [113]. The SCFA, butyrate, has also been reported to stimulate AhR activity [114]. Stimulation of AhR by these ligands is known to enhance intestinal barrier function [38,115] and reduce inflammation [116]. It is suggested that these effects are mediated via IL-22, a cytokine involved in multiple aspects of intestinal barrier function and intestinal integrity [117,118]. Recently there has been an increased focus on the concentration of tryptophan derived AhR ligands in the digesta and faeces as potential markers for improved gut health [119,120].

Interestingly, in colitis induced mice, intestinal inflammation is attenuated following inoculation with three *Lactobacillus* strains capable of metabolizing tryptophan or by treatment with an AhR agonist [121]. Similarly, I3A, an indole derivative that is produced via tryptophan metabolism by *Lactobacilli*, restores IL-22 production and ameliorates colitis when administered to colitis induced mice [117]. Supplementation with the indole derivative, indole-3-carboxaldehyde, to weaned pigs has no effect on intestinal barrier function and morphology or growth performance in the postweaning phase but increases alpha diversity in the colon and increases jejunal, ileal and colonic indexes, while also upregulating the expression of proliferating cell nuclear antigen in the intestine [122]. Supplementing cocultures of *Lactobacillus acidophilus* and *Bacillus subtilis* to postweaned pigs has beneficial effects on performance and a number of GIT health parameters, but more precisely, it leads to the upregulation of the protein expression of IL-22 and AhR [123]. The supplementation of pectin, a prebiotic substrate, to pigs in the postweaning phase increases the abundance of *Lactococcus* and *Enterococcus* abundance, increases tryptophan-metabolite concentration, increases the expression of *AHR*, *IL-22,* and other AhR-IL22 pathway genes, improves intestinal integrity and reduces inflammation [120]. These studies highlight the potential of utilizing probiotics with strong tryptophan metabolizing and AhR activating capabilities as a therapeutic strategy to enhance host GIT health. Further investigation into probiotics with these capabilities and in vivo pig studies to determine their efficiency is warranted.

#### 2.1.6. Enhancing Intestinal Immune Defence

The intestine is continuously exposed to foreign substances and its innate immune response and interaction with the adaptive immune system is crucial to ensuring optimal health. To protect the body from foreign antigens, the intestine has a number of innate defence mechanisms, including: (1) epithelial barrier integrity; (2) host defence peptides and secretory immunoglobulin A; and (3) mucus layer. The main environmental factors that can affect the functioning of these defence mechanisms, include toxins [124], pathogens [125], and stress [126]. Probiotics and their postbiotic metabolites are associated with the enhancement of multiple different aspects of the intestinal immune defence, as follows:(1)Epithelial barrier integrity

The intestinal epithelial layer serves as a barrier between the luminal environment and the body, regulating the uptake of essential nutrients and water while preventing the entry of foreign antigens into the paracellular area. The integrity and permeability of the barrier are maintained by intracellular adhesion complexes, known as tight junctions, that contain multiple transmembrane proteins. Dysregulation of these proteins can increase gut permeability, leaving the host susceptible to infection. Many pathogenic bacteria produce enterotoxins which modulate the epithelial barrier function through the manipulation of tight junctions and cytoskeletal cell components [91,92,127]. Heat stress and the stress of weaning is associated with reduced intestinal barrier integrity [128,129]. Certain probiotic strains, such as *Lactobacillus plantarum*, *Lactobacillus reuteri,* and *Lactobacillus casei*, increase tight junction gene expression and protein abundance and thereby enhance intestinal barrier function [130,131,132].

Transepithelial electrical resistance (TEER) is a measurable parameter associated with barrier function of in vitro models [133]. Probiotic strains have been demonstrated to maintain and improve TEER [131,134]. Importantly, probiotics help to maintain TEER in the presence of pathogenic bacteria by enhancing the expression of tight junction proteins [131,135,136]. This is also evident in vivo; *Lactobacillus casei* supplementation increased tight junction proteins and was highlighted as an effective preventative treatment against the adverse effects of *Escherichia coli*, specifically its effects on tight junction proteins [132]. It has been suggested that the mechanism for improved barrier function is mediated by signalling via the pattern receptor, Toll-like receptor-2 (TLR2) [137,138]. Gastric infusion of SCFA increases mRNA abundance of Occludin (*OCLN*) and Claudin-1 (*CLDN1*) in weaned piglets, highlighting SCFA as the key mechanism through which probiotics improve barrier function [37], and as discussed previously, AhR ligands may also contribute [38]. These studies highlight the cumulative effect of probiotics and their postbiotics on improving gut barrier function.

(2)Host defence peptides and secretory immunoglobulin A

Host defence peptides (HDPs) are antimicrobial peptides that are part of the innate immune system, produced in various cells and tissues of the body [139]. There are two major categories of HDPs in pigs: defensins and cathelicidins [140]. Enhancing the secretion of HDPs can improve the pig’s intestinal innate immune response due to their antimicrobial activity [141,142]. Several probiotics, including multiple strains of *Lactobacillus*, increase the expression of HDPs without triggering an inflammatory response [130,143,144,145,146]. To date, the exact mode of action is unclear but with regard to *Lactobacillus plantarum* it is suggested that TLR2 is involved in the subsequent upregulation of HDPs [145]. The increased production of SCFA associated with probiotic supplementation may also play a role [147]. The SCFA, especially butyrate, are known to increase the secretion of HDPs [103]. Another secretory product that plays a role in mucosal defence is secretory IgA (sIgA). In the intestine, sIgA is secreted into the mucus where it binds to pathogens and toxins, preventing their adhesion to the epithelium in a process referred to as immune exclusion, while also promoting immune tolerance and gut homeostasis (as reviewed in [148]). Interestingly, supplementation with probiotic *Lactobacilli* strains leads to an increase in sIgA in piglet faeces, indicating that the probiotic enhances the secretion of sIgA in the intestine [149,150,151].

(3)Mucus layer

Goblet cells are specialized secretory cells that populate the GIT and are responsible for the production of the mucin glycoproteins as the principal component of the protective mucus barrier. In the small intestine there is a single unattached “loose” mucus layer; however, in the colon there are two distinct layers of mucus, comprised of an attached “thick” inner layer and unattached “loose” outer layer [152,153]. The single mucus layer of the small intestine enables absorption of nutrients while limiting bacterial contact with the epithelium [153,154,155]. In the large intestine, the inner layer is impermeable to bacteria, preventing them from overactivating the immune sensing and responsive cells in this region and triggering inflammation [152,156,157]. In contrast, the outer mucus layer of the colon is a habitat for microbes, called mucus-associated microbes which utilize the mucin glycans as attachment sites [158,159].

Some pathogens secrete enzymes that cleave the mucins and this is a key aspect to their pathogenicity [160]. Related to this, other pathogenic microbes rely on the breakdown of the mucus layer by other bacteria to proliferate and then opportunistically infect the host [161]. The mucus layer can become impaired during periods of intestinal dysfunction, such as at weaning time, increasing the risk of exposure to pathogenic microorganisms [162]. To this extent, stabilizing the mucus layer by promoting optimal goblet cell numbers/activity and mucus secretion is beneficial to host health and can potentially reduce the risk of infection by pathogens. Probiotic supplementation to pigs has been shown to enhance goblet cell numbers and mucus expression in the intestine [163,164,165]. For example, supplementing two week old pigs with a blend of lactic acid bacteria and yeast increases the numbers of goblet cells in the large intestine [166]. Pigs challenged with ETEC have reduced goblet cells numbers in the intestine; however, pretreatment with *Bacillus licheniformis* and *Bacillus subtilis* offsets this decrease in ETEC challenged pigs [164]. Pretreatment with a moderate dose of *Bacillus licheniformis* and *Bacillus subtilis* increases *MUC2* expression in the intestine of both ETEC challenged pigs and unchallenged control pigs [164]. These studies highlight the potential for probiotics to enhance goblet cell numbers and mucin secretion, thereby stabilizing the mucus layer of the intestine.

#### 2.1.7. Reducing Inflammation

Cytokines are small signalling proteins produced by a wide range of immune and intestinal epithelial cells. Cytokines can be pro- or anti-inflammatory and play a key regulatory role in the immune and inflammatory response. A balance between these classes of cytokines is a crucial aspect to obtaining a regulated immune response in the gut, as uncontrolled inflammation has negative effects on the integrity and functioning of the GIT barrier [167]. In particular, increased and unregulated production of proinflammatory cytokines such as TNF-α, IFN-γ, and IL-6 are associated with reduced barrier integrity [131,168]. A number of probiotic species have been shown to restore the balance of inflammation in in vitro models of the gut. The addition of *Lactobacillus reuteri* can offset stimulated inflammation by maintaining barrier integrity in an LPS-stimulated porcine jejunal epithelial cell line (IPEC-J2) and also reduced the expression of both TNF-α (*TNF*) and IL-6 (*IL6*) [131]. Pretreatment of intestinal porcine cells with the probiotic *Lactobacillus plantarum* decreases the expression of pro-inflammatory cytokines IL-8 (*CXCL8*) and TNF-α (*TNF*) when stimulated with ETEC K88 [125]. *Lactobacillus amylovorus* suppresses TLR4 signalling caused by *Escherichia coli* [169]. These studies suggest that many probiotics can rebalance the overexpression of pro-inflammatory cytokines caused by pathogenic immune stimulation.

#### 2.1.8. Reducing Oxidative Stress

Oxidative stress is caused by an imbalance between oxidants and antioxidants in favour of oxidants, also referred to as reactive oxygen species (ROS), which in turn can lead to cell and tissue damage [170]. There are a number of developmental stages including gestation, lactation, and weaning where the pig is normally subjected to oxidative stress. Environmental factors including mycotoxins in feed, social change and disease can also induce oxidative stress [171]. Oxidative stress is typically accompanied by a reduction in average daily gain (ADG) and a reduction in average daily feed intake (ADFI) [172]. Intestinal oxidative stress can reduce GIT barrier integrity and dysregulate cell homeostasis leading to erosion of the intestinal villi and significant losses in intestinal enzyme activity and absorptive function [172]. Oxidative stress and inflammation are interconnected in a complex interplay where one can easily be induce by another (as reviewed in [173]).

Interestingly, probiotics and their postbiotic metabolites can help reduce oxidative stress and alleviate the negative issues associated with it [174,175,176]. The culture supernatant, intact cells and intracellular free extracts of *Bifibacterium animalis 01* all have strong antioxidant activity in vitro [174]. In vivo, the inclusion of multispecies probiotics including *Lactobacillus acidophilus*, *Lactobacillus casei*, *Bifidobacterium thermophilum,* and *Enterococcus faecium* to pigs in the postweaning phase reduces intestinal oxidative stress [175]. The inclusion of *Lactobacillus salivarius* or *Lactobacillus plantarum* in the diet of pigs postweaning improves the antioxidant capacity of the pig, evident from markers present in both the blood and intestine [176,177]. *Bacillus amyloliquefaciens* supplementation reduces the oxidative stress marker malondialdehyde (MDA), in the intestine of finisher pigs while increasing the expression of antioxidants enzymes [178]. *Lactobacillus casei* supplementation alleviates the spike in MDA in the liver of postweaned pigs challenged with LPS, while also increasing levels of the antioxidant enzyme, superoxide dismutase (SOD) [179]. The beneficial effects observed in these studies suggest that the potential of probiotics as therapeutic agents targeting the reduction in oxidative stress in the pig and may be particularly useful at specific timepoints, including weaning, gestation, and lactation. 

#### 2.1.9. Other Potential Beneficial Effects: Vitamins and Enzymes

Certain bacteria in the GIT microbiota, such as Bifidobacteria, supply vitamins to the host by de nova synthesis. Most water-soluble B vitamins such as vitamin B-6, vitamin B-12, biotin, riboflavin, thiamine, and vitamin K are all produced by the microbiota [180]. While the production of vitamins by the microbiota has long been established, the mechanisms of absorption and bioavailability of these vitamins to the pig are less well known. Specific transporters for the different water-soluble B vitamins have been identified in the colon of humans and mice [181,182,183,184]; however, analogous studies are limited in pigs. Branner and Roth-Maier documented aspects of synthesis and absorption of several B vitamins in grower pigs and reported that: thiamine and biotin were synthesized in the colon, but the degree of absorption was unclear as levels were greater in the faeces than precaecal digesta; riboflavin and pantothenic acid were absorbed and utilized in the colon, as there was greater levels in the precaecal digesta than in the faeces [185]. While this study confirms the synthesis of vitamins by the GIT microbiota and vitamin utilization, at least for certain vitamins, the extent of the absorption of these vitamins remains unclear. Certain bacteria utilize either dietary or B vitamins produced by other bacteria and it is suggested that there may be competition between the host and the intestinal bacteria for these B-vitamins [186]. The production of vitamin K and B vitamins is often cited in the literature as a beneficial feature of the microbiota; however, the extent of the availability and utilization of these vitamins in pigs has not been established fully.

In addition to vitamin production, probiotics, such as *Bacillus* and Lactobacillus strains, produce extracellular enzymes that break down undigested food in the GIT [187,188,189]. Hence, with regard to pig nutrition, additional nutrients are made available in this way, contributing to improved feed utilization [190]. 

### 2.2. Effects of Probiotic Inclusion in the Maternal Sow Diet on Sow and Offspring Microbiota, Health and Performance

The beneficial effects of probiotic supplementation have been researched extensively over the past number of years, and the importance of a balanced and diverse microbiota has become increasingly evident. Mechanisms for the beneficial effects of probiotics, particularly in the maternal diet, have, however, yet to be ascertained. This review has so far focused on the modes of action of probiotics. The focus will now be on why maternal probiotic supplementation, in particular, is an area with immense potential. Maternal probiotic supplementation enables both the sow and her offspring to benefit as maternal supplementation transmits the probiotic to the offspring (via sow faeces or milk), acting as an indirect form of probiotic supplementation to the offspring [2]. In addition, the beneficial effects of maternal probiotic supplementation are timely during this critical period, affording an opportunity to not only enhance sow health but also foetal and early postnatal health extending to the lifetime health of the offspring.

#### 2.2.1. Early Life Gut Colonization—Plasticity of the Pioneer Microbiota

Promoting a healthy microbiota is essential at all stages of the pig’s life; however, the early stage of life presents a window of opportunity to modulate the microbiota and improve animal health and performance due to the high plasticity of the young piglets’ GIT microbiota [2,191]. Rapid colonization of the piglets GIT commences immediately after birth, with the initial colonizing population termed the “pioneer microbiota”. The pioneer microbiota plays a fundamental role in the development of the microbial communities that persist later in life as it initiates a chain of microbial succession, highlighting the immediate postnatal period as a critical window of GIT microbiota assembly [192,193,194].

Studies in pigs indicate that some colonization of the intestine occurs prior to birth [195,196,197], although predominant colonization occurs during the postnatal period. In a recent study by Wang et al., maternal supplementation with *Lactobacillus reuteri* increased the alpha diversity and enriched the abundance of twenty-one bacterial taxa at genus level in the meconium of newborn piglets [198]. This study confirms that maternal GIT bacteria are vertically transmitted to the foetus and that maternal probiotic supplementation can be utilized as a tool to alter the composition of this bacterial exposure. The influence of the prenatal microbiota on the establishment of the pioneer microbiota remains to be answered; however, this prenatal microbiota may affect the development of the immune system, which will be discussed further under the title “Offspring Immune System Development and Programming” of this review.

The importance of the pioneer microbiota, in combination with the high adaptability and plasticity of the young piglets’ GIT microbiota, emphasizes how early modulation is a worthy area of exploration with regards to promoting enhanced health and development. An interesting study conducted by Chen et al. set out to determine the main environmental contributors to the development of the microbiota in newborn pigs and concluded that the sows’ milk, the sows’ nipples, and the slatted floor were the most likely early sources of microbes; however, throughout lactation, the piglets acquire GIT ecosystems that primarily imitated that of their mothers rather than the housing environment [199]. In support of this, when sows are supplemented with probiotic bacteria that is typically not present in the faeces, the same strain can later be detected in both the sow and her offspring’s faeces [2]. Interestingly, the microbiota of the sows faeces shifts in composition during gestation and lactation, with early lactation (day 3) exhibiting the lowest bacterial richness and diversity level and an increase in potentially pathogenic bacteria *Proteobacteria* and *Fusobacteria* [200]. These studies underline the potential scope for improving the composition and diversity of sow’s microbiota to enhance the offspring’s microbiota.

Furthermore, the sows GIT microbiota may be a source of bacteria present in the mammary gland and milk, via the entero-mammary pathway, suggesting that modulation of the GIT microbiota may have an influence on the milk microbiota [201]. This phenomenon is partly supported by the finding that maternal supplementation of *Bacillus altitudinis* to sows during late gestation and lactation alters the composition and reduces the alpha diversity of the colostral microbiota [20]. However, this reported change in bacterial composition in the colostrum may be due to alterations in the nutrient composition that were also noted with *Bacillus altitudinis* supplementation [20]. Similarly, supplementation of *Lactobacillus reuteri*, in the sows diet during gestation, alters the colostral microbiota [198]. In rats, maternal supplementation of *Lactobacillus plantarum* increases the alpha diversity of the milk microbiota [202]. Greiner et al. investigated the existence of this entero-mammary pathway in sows and concluded that the observations could support, but not prove, its existence in sows [203]. The colostrum microbiota and its diversity are a relatively recent area of exploration; its precise influence on the developing offspring remains relatively elusive as it is confounded by so many other factors. These studies all highlight the sow as the major contributor to the establishment of the offspring’s microbiota, and that maternal probiotic supplementation could therefore influence the microbiota later in life and potentially improve the lifetime performance of the offspring [2,20].

The abrupt weaning practice in commercial systems promotes a greater incidence of intestinal dysbiosis, presenting an opportunity for enteric pathogens such as *Escherichia coli* and rotaviruses to proliferate [8,9]. To reiterate, the relative abundance of specific groups in the suckling pig’s microbiota, such as *Lactobacillaceae*, *Ruminococcaceae*, *Lachnospiraceae,* and *Prevotellaceae,* are linked to a reduced incidence of diarrhoea in the pig postweaning [10]. Oral inoculation of preweaned pigs with a probiotic blend improved their ability to clear ETEC when challenged postweaning [204]. Preweaning supplementation with *Lactobacillus johnsonii* or *Bacillus subtilis* reduces diarrhoea in pigs postweaning [149]. While maternal probiotic supplementation, with *Bacillus altitudinis*, has a greater impact on the offspring’s intestinal microbiota on day 8 postweaning compared to direct probiotic supplementation postweaning [20]. Moreover, in the same study by Rattigan et al., maternal probiotic supplementation increased the abundance of *Lactobacillus* in the offspring faeces on day 118 postweaning [20], supporting the theory that the pioneer microbiota influences the microbial communities later in the pigs life. In contrast, Beaumont et al. concluded that the early life microbiota is not a major factor underlying the susceptibility to diarrhoea in the postweaned pig [205]. These studies suggest that modulation of the offspring’s microbiota preweaning may be an effective therapeutic strategy to reduce the proliferation of pathogens and prevalence of diarrhoea in the offspring postweaning. In combination, the literature indicates that modulation of the pioneer microbiota is a worthy therapeutic target, and to effectively modulate its establishment the focus must be on the sow’s microbiota, at least in late gestation and lactation. However, further analyses of the preweaned pig’s microbiota and its link to postweaning diarrhoea susceptibility would help add to the current knowledge and understanding.

#### 2.2.2. Oxidative Stress

In sows, there is a notable increase in systemic oxidative stress in the final month of gestation and during lactation [206,207]. The significant foetal development and mammary growth that occurs during the final month of pregnancy, followed by milk production during lactation, increase the metabolic burden on the sow, which then also increases systemic oxidative stress [206,208]. During gestation, oxidative stress can cause adverse effects such as reduced feed intake, hindered foetal development, and an increase in stillbirth rate [209,210]. Similarly, during lactation, oxidative stress can reduce feed intake and subsequently reduce colostrum and milk yield, all of which have negative effects on litter performance, reducing piglet and litter weights at weaning [211,212,213]. The sow’s litter performance during lactation can be related to both the sow’s GIT microbial composition and oxidative stress status [214]. Furthermore, oxidative stress can negatively affect the sows reproductive performance by inhibiting oocyte maturation, fertilization, and increasing perinatal mortality [215]. Sows with large litter sizes (>14 piglets) have increased oxidative stress markers and decreased antioxidant capacity in late gestation and lactation [216]. With the current average litter size greater than 14 piglets per sow [217] and rising yearly, interventions to alleviate sow oxidative stress are becoming increasingly important.

For the young pig, birth and weaning represent timepoints that are associated with increased oxidative stress [218,219,220]. At birth, the pig is exposed to a complete change in environment and diet which can lead to the production of large amounts of ROS [218]. Increased ROS, combined with the poorly developed antioxidant capacity, drives a state of oxidative stress in the newborn piglets [218] and in the postweaned pig [219,221]. Intestinal oxidative stress can cause a reduction in GIT barrier integrity and dysregulation of cell homeostasis leading to erosion of the intestinal villi which results in dramatic losses in intestinal enzyme activity and absorptive function [172].

Several probiotic strains have been beneficial in reducing oxidative stress and enhancing the antioxidant capacity in the pig [174,175,176,177,178,179]. In this regard, maternal probiotic supplementation provides an effective approach as it can potentially reduce oxidative stress in both the sow during late gestation and lactation and the offspring at birth and weaning time. Supplementation with probiotics enhances the sow’s antioxidant capacity and reduces oxidative stress during gestation and lactation, promoting improved sow health and reproductive performance parameters [29,222,223].

This reduction in oxidative stress during gestation can subsequently improve oxidative status in the offspring by: (1) improving the antioxidant capacity of the prenatal pig, thus improving the animals ability to buffer the spike in free radicals that occurs at birth [224]; (2) improving antioxidant levels in the colostrum and milk which can reduce oxidative stress in the piglet [211,225]; (3) potential inoculation with the probiotic bacteria that have oxidative stress alleviating capabilities, affording the growing piglet longterm health benefits. This approach is supported by observations that the sow’s antioxidant capacity is correlated to the offspring’s antioxidant capacity and oxidative stress status [15,29,211,226]. Oral probiotic supplementation to the pig preweaning enhances oxidative status in the postweaning phase, underlining the potential for preweaning microbiota modulation to reduce postweaning oxidative stress [227]. These studies highlight the potential multipronged action of maternal probiotic supplementation as a means of reducing oxidative stress in both the sow and her piglets, promoting improved foetal development, sow lactation performance and health and development of the offspring [15,29,223].

#### 2.2.3. Inflammation

In sows, parturition and the immediate postparturition period is associated with the presence of systemic inflammation, which can potentially cause postpartum dysgalactia syndrome (PDS) [228,229]. Limiting this inflammation can in many instances reduce the risk of inflammatory induced physiological and behavioural changes which would adversely affect the sow’s performance and that of her offspring. To this extent, there have been a number of studies investigating the use of anti-inflammatory drugs postpartum, although results of these studies have been variable in outcome [230,231,232,233,234].

Modulating the sow’s microbiota via dietary intervention is an alternative technique that can potentially reduce inflammation and improve sow health and performance around the time of parturition. Increasing dietary fibre alters the sows microbiota and also reduces both intestinal and systemic inflammation, highlighting the GIT microbiota as a potential target for reducing systemic inflammation in the sow [226]. To date, probiotic supplementation to sows during gestation and lactation has returned variable results in terms of serum inflammatory markers. Laskowksa et al., observed an increase in both serum inflammatory and anti-inflammatory cytokine concentrations prepartum in sows supplemented with the probiotic blend “Bokashi^®^” [235]. While no difference in sow serum inflammatory markers were reported in an analysis of the effect of the supplementation with different probiotic strains or synbiotic combinations [236]. In dairy cows, supplementing *Bacillus subtills* reduces the occurrence of mastitis, which is a mammary gland inflammatory disease [237]. The potential of probiotics to reduce inflammation exists; however, further investigation is required to determine if they are an effective strategy, and more precisely what specific probiotic strains are most effective, to reduce systemic inflammation in sows around parturition.

#### 2.2.4. Offspring Immune System Development and Programming

Late gestation and early life present the earliest window of opportunity for the microbiota and its metabolites to influence the development and programming of the immune system. Although a relatively new research topic, there is increasing evidence of a role of the maternal GIT microbiota and its metabolites on the development and programming of the foetal immune system [238,239,240,241,242]. Microbes and their bioactive metabolites may be directly transmitted from the maternal microbiota to the foetus [198,240], and can exert influence on immune development [239]. However, functional differences in the placenta between species affect transfer characteristics [243], and findings from studies on one species cannot be immediately generalized to another. Bacterial metabolites, such as SCFA and AhR ligands, have been reported to epigenetically influence or “imprint” the foetal immune system [239,241,242]. Through the inhibition of histone deacetylases (HDACs), SCFA influences the development of Treg cells, via *Foxp3* expression, in the murine foetal lung, thereby reducing the severity of induced allergic airway disease in adult offspring [239]. The AhR ligand, indole-3- carbinole, activates AhR in the foetus and prevents development of the intestinal disease necrotizing enterocolitis in the offspring of mice [242]. While maternal supplementation with *Lactobacillus rhamnosus* in mice alters placental expression of proinflammatory cytokines, reducing *IL4* and increasing *TNF* expression levels, which is associated with reduced allergic airway inflammation in the offspring [241]. 

The pioneer microbiota and early life microbial exposure play an important role in subsequent immune adaptations and functioning of the immune system later in life [244]. This process is often referred to as “immunological imprinting” (as reviewed in [245]). Treating pigs with antibiotics from day 1 to 14 of life drives a more rapid and pronounced proinflammatory response when pigs are subsequently challenged at day 49 of life [246]. Furthermore, early-life antibiotic treatment effects the microbiota at day 176 and intestinal gene expression at day 55 after birth [247]. In contrast to antibiotic treatment, supplementation with *Lactobacillus reuteri* in the neonatal phase promotes the development of the intestinal immune system by enhancing Peyer’s patches development, increasing mucin and antimicrobial peptide expression, and increasing the number of CD3^+^ cells in the ileum or the jejunum [146]. Interestingly, supplementation of *Lactobacillus reuteri* in the maternal sow diet alters the metabolite profile present in the blood of the umbilical cord, thereby altering the metabolite profile that the developing foetus is exposed to [198]. Although it is clear that the microbiota has an important role in immune system development and adaptation, the exact nature of these immune–microbe interactions in neonates remains elusive, particularly in terms of how maternal probiotic supplementation might influence the offspring’s immune system imprinting. The effect of probiotic supplementation on the establishment of microbiota and subsequent microbial succession makes it difficult to distinguish between effects influenced due to neonatal immune system imprinting and effects influenced by the GIT microbiota.

#### 2.2.5. Enhancing the Immune Potential of the Colostrum and Milk

Colostrum is the first milk produced after birth and consists predominantly of carbohydrates, lipids, and proteins with small quantities of minerals, vitamins, leukocytes, and somatic cells [248]. The colostrum is initially rich in immunoglobulins, with a marked drop in concentrations over the first 24–48 h of lactation [249]. The placentae of many domestic species, including pigs, are impermeable to immunoglobulins and therefore offspring only acquire passive immunity from colostrum [250]. Intriguingly, in the newborn pig, intestinal cells can absorb immunoglobulins intact and transport them into the blood stream. However, this ability ceases within the initial 24–48 h of life, in a process known as “intestinal closure” [251,252]. These immunoglobulins are a vital source of passive immunity for the immunologically naïve piglet in the initial weeks of life and play a significant role in early postnatal development of the mucosal and systemic immunity of the piglet [253,254]. 

Piglet colostrum intake has direct implications on preweaning mortality [255] and bodyweight gain [256]. Not surprisingly, piglet plasma immunoglobulin content at weaning time is related to both plasma immunoglobulin content 24 h after birth and to colostrum intake [253,255]. The drastic increase in litter sizes over the past decade [217] has led to increased competition for colostrum and subsequently reduced intakes, which has negative consequences for the piglets’ health and performance [255,256,257]. Improving the quality of colostrum and milk has the potential to enhance the health and performance of the offspring, as evident in the improved performance and immune response of gilt offspring when they are supplemented with sow colostrum, which is known to have enhanced immune potential [258,259]. 

In regard to enhancing the immune potential of colostrum and milk, there are several components which can potentially be modulated through maternal probiotic supplementation. Increasing the immunoglobulin content of the colostrum and milk can increase the passive immunity transferred from the sow to her offspring and thereby enhance offspring performance. A number of different probiotic strains have successfully increased the immunoglobulin content of the sow’s colostrum and milk when included in the maternal diet [260,261,262,263]. Furthermore, oral supplementation of a probiotic blend, containing *Bacillus mesentericus*, *Clostridium butyricum,* and *Enterococcus faecalis*, in combination with a vaccine injection to sows, increased the concentrations of IgA, IgG, and more interestingly, increased vaccine-specific antibodies in the sow serum and in the colostrum of sows compared to vaccinated sows that did not receive the probiotic [263]. This affect is suggested to be because of immune stimulation by the probiotic. These studies highlight the how maternal probiotic supplementation can stimulate the sow’s immune system and thereby enhance the immunoglobulin content of the colostrum. 

In terms of the immunological capacity of colostrum, the immunoglobulin content is typically referred to; however, other immune modulating bioactives, such as antioxidants, microbes, cytokines, leukocytes, oligosaccharides, and antimicrobial peptides are also present [264,265,266,267]. Unlike the limitations of utilizing studies from different species on placental transfer, this form of postbirth immune system modulation may have more parallels with human findings. As alluded to previously, cytokines are small immune signalling proteins. The concentration of cytokines is typically greater in the colostrum compared to milk [268]. Interestingly, the cytokine profile of the colostrum and milk often reflects the health of the mother [269,270,271]. The subsequent effects on the offspring of the variations in colostrum and milk cytokine and if the greater levels of pro-inflammatory cytokines predispose the offspring to an inflammatory condition has yet to be alluded to. The cytokines in the colostrum and milk are suggested to play an essential role in the offspring as both mediators of early immune responses to antigens and influencers of immune system development [268,272,273].

In humans, breast milk from mothers with inflammatory bowel disease has increased pro-inflammatory cytokines compared to healthy mothers [269]. Furthermore, in humans, greater milk IL-6 is associated with lower relative weight, weight gain, percent fat, and fat mass at 1-month of age [274]. Subclinical mastitis in humans is associated with a pro-inflammatory/Th-1 cytokine predominant profile in the milk [270]. Similar studies investigating the variation in cytokine profile in the milk between healthy and nonhealthy sows would provide valuable knowledge in this area. Interestingly, sow colostrum and serum contain a greater concentration of the cytokines GM-CSF, IFNγ, IL-1α, IL-1RA, IL-2, IL-4, IL-6, IL-10, IL-12, IL-18, and TNFα at 3 h postfarrowing compared to gilts [259]. Supplementing the sow with a probiotic blend (Bokashi^®^) increases the concentrations of pro-inflammatory cytokines, TNF-α and IL-6, and anti-inflammatory cytokines IL-4, IL-10, and TGF-β in the colostrum and milk [235]. The increase in pro-inflammatory cytokines is suggested to increase the protective capacity of the colostrum by generally stimulating the immune system, while the increase in cytokines, such as IL-10 and TGF-β, have a role in immune regulation through the promotion of Treg differentiation [235]. In humans, maternal supplementation of a probiotic blend containing *Lactobacillus casei*, *Bifidobacterium longum*, and *Bacillus coagulans* increases the level of the anti-inflammatory cytokine IL-10 in the milk [275]. The capacity for modulation of the colostrum and milk cytokine content by probiotics is evident. However, there is a need for innovative in-depth studies to analyse the effects of cytokines in the colostrum and milk on the development of the offspring. 

Colostrum contains leukocytes, such as macrophages, B cells, granulocytes, and various T lymphocyte subsets [276] and while knowledge of their precise role is limited [277], it is believed that they are important in supporting the immune system of the offspring while it is developing [278]. The leukocytes from the colostrum are absorbed through the digestive tract and then into the blood by the offspring [279,280]. A subset of the T-cells present in the colostrum are suggested to be effector memory cells [281,282]. In this regard, given that colostral T cells are absorbed into the blood of the offspring [279,280], the effector memory T cells in the colostrum are another form of passive immunity transferred from the sow to her offspring. This is supported by observations that functional antigen-specific T cells are transferred from vaccinated sows to the offspring, via the colostrum, and participate in the neonatal immune response upon stimulation [283]. 

Interestingly, the concentration and the ratio of leukocytes subsets can vary greatly between gilts and sows, with sow colostrum having a much greater concentration of leukocytes [276]. Despite the significant effect of probiotics on the immune system, to date, there are a limited number of studies investigating the effect of maternal probiotic supplementation on leukocyte content of the colostrum and milk in sows. An improvement in immunological quality of the colostrum and milk was observed, with increases in subpopulations of B cells with CD19+, CD21+, and CD5+CD19+ expression, in sows supplemented with the probiotic preparation EM Bokashi^®^ [262]. However, a reduction in the percentage of cells expressing membranous CD14 was observed in sows milk following maternal supplementation with *Enterococcus faecium* [284]. In the same study, the number of CD14^+^ milk cells was positively correlated with the percentage B cells and activated T cells in the ileal mesenteric lymph nodes of offspring [284]. These studies indicate the potential of probiotics to modulate the leukocyte component of the colostrum and milk, and subsequently influence the offspring; however, future studies are warranted to contribute to a better understanding of the relationship between maternal probiotic supplementation and colostrum and milk leucocyte composition.

The presence of microbial metabolites in milk, their role in offspring development, and the potential influence of the maternal microbiota on milk composition is an area where research is presently sparce. Stinson et al. recently reviewed the limited literature on the various possible origins for metabolites in human breast milk, suggesting both local production in the mammary gland, by the milk microbiota, and systemic circulation, of metabolites originally produced by the GIT microbiota, as possible origins the microbiota metabolite present in milk [285]. Supplementation of *Lactobacillus reuteri*, in the maternal sow diet during gestation, increases the concentration of lactate in the sow’s milk [198]. The beneficial effects of certain microbiota metabolites and the potential to enhance their presence in milk, through modulation of the maternal microbiota illuminate this as an intriguing area for future exploration. 

#### 2.2.6. Sow Feed Intake and Digestibility

Lactation is a metabolically demanding period during which sow feed intake must be optimized to ensure adequate milk production and to reduce the impact on sow weight and body condition [286]. Inadequate feed intake during lactation negatively affects sow body condition, piglet weaning weight, time taken to return to oestrus, and subsequent farrowing rates [287,288]. Ensuring high feed intake during lactation is crucial, particularly in recent years, as the average litter sizes have increased [217], with subsequent increases in the demand for milk from the sow. Probiotic supplementation to sows increases lactation feed intakes possibly due to the enhanced health and nutrient uptake of the sow [289,290,291,292,293,294]. Additionally, supplementation with *Enterococcus faecium* or *Bacillus* strain probiotics improves nutrient digestibility in sows [48,295]. Therefore, it can be speculated that through enhanced feed intake, utilization, and nutrient absorption, probiotics can reduce sow weight loss during lactation, improve sow milk yield and composition, and enhance reproductive performance of the sow. In support of this, several studies have reported that probiotic supplementation, such as *Bacillus* strains, yeast strains, and probiotic blends, enhanced milk yield, and improved milk composition, such as increases in the fat, protein, and lactose content, although results have been highly variable in this regard [2,55,260,289,290,294]. Furthermore, inclusion of *Bacillus subtilis* or *Saccharomyces cerevisiae* in the sow diet can reduce the wean to service interval and improve future reproductive performance of the sow [261,290,292]. These sow performance benefits are likely due to the combination of multiple modes of action resulting in improved sow health, feed intake and utilization and thereby enhanced sow performance, and subsequently litter health and performance. Table 2 includes a summary of the effects of probiotic supplementation on the sow and offspring microbiota and other noteworthy effects observed.

## 3. Synbiotics

Synbiotics are defined as “a mixture comprising live microorganisms and substrate(s) selectively utilized by host microorganisms that confers a health benefit on the host” [305]. A common misconception with synbiotics is that the ‘syn’ prefix in the word ‘synbiotic’ implies synergy; however, it means ‘together’ [305]. To clarify this matter, in 2019, the International Scientific Association for Probiotics and Prebiotics (ISAPP) defined two subsets of synbiotics: complimentary synbiotics and synergistic synbiotics [305]. A complementary synbiotic is defined as “a synbiotic composed of a probiotic combined with a prebiotic, which is designed to target autochthonous microorganisms” [305]. A synergistic synbiotic is defined as “a synbiotic in which the substrate is designed to be selectively utilized by the co-administered microorganism“ [305]. An important distinction between the two synbiotic types is that a synergistic combination must provide greater beneficial effects than the prebiotic or probiotic alone, whereas a complimentary synbiotic need only provide a beneficial effect compared to a control [306]. To date there have been a limited number of studies investigating the use of synbiotics in the maternal sow diet, and of those relatively few have compared the effects to probiotic supplementation alone and none have assessed for synergistic effects. 

Prebiotics consist of a broad range of substrates and have been used to varying degrees of success in pigs (as reviewed in [307]). Here, the focus is to discuss the potential role of synbiotics in the maternal sow diet and briefly detail some additional benefits of the inclusion of a prebiotic in combination with probiotics in the maternal sow diet. Recently, the definition of a prebiotic was updated, expanding the categorization to not only include nondigestible carbohydrates, but also include novel prebiotics such as amino acids, peptides, and nucleotides (as reviewed in [308]). This update in definition highlights the potential of investigating combinations of novel prebiotics and probiotics to identify new complementary and synergistic combinations, which may amplify the beneficial effects observed from probiotics supplementation alone. Certain prebiotics have microbiota-independent immunomodulatory effects; nondigestible oligosaccharides, for example, can directly modulate the inflammatory response via interaction with TLR4 [309,310], and beta-glucans are also capable of directly modulating the immune system (as reviewed in [311,312]). The immunoregulatory properties of certain prebiotics suggest that maternal supplementation with a prebiotic in combination with a probiotic could bolster the capacity of the supplement to enhance immune system related processes, compared to probiotic supplementation on its own. 

Another overlooked benefit of maternal prebiotic supplementation, more specifically nondigestible carbohydrate prebiotics, is its ability to modulate the colostrum and milk oligosaccharide content. The oligosaccharide content in the colostrum and milk acts as a prebiotic substrate for bacteria in the suckling pigs GIT, and thus has an impact on shaping the GIT microbiota in early life. This is supported by evidence that there is minimal intact oligosaccharides present in the faeces of 1–2 d old pigs, suggesting that they are fermented by bacteria in the neonate, even in the initial hours of life [313]. The oligosaccharides in the sows colostrum and milk are a group of complex oligosaccharides referred to as porcine milk oligosaccharides (PMOs) and aside from providing substrates to beneficial bacteria in the GIT, PMOs can also influence the intestinal immune system and pathogen proliferation [314,315,316,317]. PMOs exert beneficial effects on the intestinal immune system by reducing inflammation caused by pathogens. This has been observed both in vitro, using intestinal epithelial cell strains and HeLa cells with human milk oligosaccharides [314], and in an ex vivo model, using human foetal intestine explants with milk oligosaccharides derived from colostrum [315]. PMOs have also been shown to improve intestinal barrier function [316] and can help reduce the attachment of pathogens to the intestinal epithelium as they act as soluble receptor analogues that compete for bacterial binding on the intestinal epithelium [317].

To date, there has been more than ninety PMO structures identified, and they can be classed into three major types: fucosylated, sialylated, and nonfucosylated [313,318,319]. PMOs comprise a lactose core linked via different bonds to N-acetylglucosamine or N-acetyl galactosamine. Additionally, fucosylated and sialylated PMOs have a fucose or sialic acid residues at the terminal positions. The relative abundance of the PMO content of the sows colostrum varies among and within breeds [320]. The total concentration of PMOs is considerably greater in colostrum compared to milk [267,319]. However, different individual PMO structural types can have different patterns of concentration across lactation, with a small number of PMO structures increasing in concentration [313,319]. Interestingly, the decrease in concentration of PMOs from colostrum to mature milk is greater in gilts than in sows [318]. This decrease is associated with a doubled rate decrease in sialylated PMO structures from colostrum to mature milk in gilts compared to sows, while the rate of decrease in neutral and fucosylated structures remains the same [318]. The bioactivity of PMOs is closely linked to their structure (as reviewed in [321]). Maternal chito-oligosaccharide supplementation modified the PMOs present in the sows milk, with increases observed for some PMOs and decreases observed for others [322]. Trevesi et al., characterized the colostrum PMO content across different breeds of sows and correlated this data with sow maternal traits [320]. Four PMO cluster groups were subsequently generated in a principal component analysis which demonstrated that cluster membership was associated with litter weight gain within first 3 days of life and at weaning, but not with piglet mortality within the initial 3 days [320]. To date, there are a limited number of studies investigating the effects of maternal dietary oligosaccharide supplementation on the PMO content of the colostrum and milk of sows. Further investigations aimed at identifying the effect of dietary maternal oligosaccharides on the concentration and structures of PMOs in the colostrum and milk and the effect this has on the offspring’s microbiota is warranted.

Currently, research investigating synbiotic supplementation to sows is limited and has focused on the combination of traditional prebiotics, such as nondigestible carbohydrates, and probiotics. In the coming years, it is realistic to anticipate an increase in research involving synbiotic combinations of probiotics and the newly classed novel prebiotics, such as microbiota modulating amino acids or “Aminobiotics” [308]. Many beneficial bacteria require amino acids for their growth and proliferation [323,324,325]. Given the high degree of protein digestion and absorption that occurs in the small intestine, amino acids can be in scarce supply in the distal intestine, limiting the growth of some beneficial strains. For growth, certain *Bifidobacterium* strains require cysteine [323], and certain *Lactobacillus* strains require arginine, lysine, and glutamic acid [324,325]. The role of amino acids as a potential prebiotics is discussed in [307].

Tryptophan is an amino acid of particular interest, not only for its microbiota modulating capabilities, but due to the tryptophan metabolites produced by the microbiota and their effect on the AhR pathway and intestinal homeostasis (as reviewed in [111,113]). Increased tryptophan supplementation to weaned pigs can increase microbiota diversity and AhR ligand concentration, enhance tight junction protein and host defence peptide expression, and reduce pro-inflammatory cytokine expression [326,327]. In mice, supplementation of a synergistic synbiotic comprise tryptophan and a *Lactobacillus reuteri* strain, with a high capacity to produce AhR ligands, had a synergistic effect on AhR activity, improving villus height: crypt depth ratios and decreasing intraepithelial lymphocyte counts [110]. Furthermore, maternal supplementation of an AhR ligand enriched diet in mice increases the presence of AhR ligands in both the amniotic fluid and the milk, leading to a reduction in the severity of necrotizing enterocolitis when induced [242]. Supplementing the combination of tryptophan with *Escherichia coli Nissle 1917*, but neither alone, reduced diarrhoea, upregulated immunoregulatory cytokines and downregulated proinflammatory cytokine in gnotobiotic pigs transplanted with human faecal microbiota and challenged with rotavirus [328]. Citrulline is another amino acid which has prebiotic characteristics. In vitro, treatment with a combination of *Lactobacillus helveticas* and citrulline enhances the growth of IPEC-J2 cells and inhibits the adhesion of *Escherichia coli* NFM138 greater than citrulline or *Lactobacillus helveticas* alone [329]. Furthermore, the addition of citrulline increases the adhesion of *Lactobacillus helveticas* to IPEC-J2 cells [329]. Citrulline supplementation to finisher pigs modulates the microbiota and increases the alpha diversity of the faeces [330]. These studies highlight the potential of citrulline to act as a prebiotic and moreover, the potential synergistic effect of a combination of *Lactobacillus helveticas* and citrulline. The role of dietary amino acids in shaping the microbiota is an area that is likely to receive increased attention in the coming years, especially with the current interest in reducing dietary crude protein to potentially improve postweaning gut health [331] and sustainability of pig production [332]. With this increased attention, additional amino acid and probiotic synbiotic combinations are sure to be identified.

Alternative novel prebiotics, such as nucleotides, may also present as intriguing components in synbiotic preparations for future explorations [307,333]. Although, in a recent study in calves, the supplementation of a combination of *Bacillus* probiotics and nucleotides in milk replacer had no synergistic impact on the growth, health or on faecal bacteria counts of *Lactobacillus* spp., *Bifidobacterium* spp., *Escherichia coli,* or *Clostridium perfringens* [334].

The potential for synbiotics is evident; however, further research is warranted to identify genuine complementary or synergistic synbiotics. In the study by Wang et al., there was no enhanced effects of maternal synbiotic supplementation compared to probiotic supplementation on offspring performance postweaning [15]. In the study by Zhu et al., maternal synbiotic and probiotic supplementation both had positive effects on the microbiota, but it is difficult to determine if either was more effective than the other [301]. It is likely that in the coming years there will be an increase in the number of synergistic synbiotics identified for use in pig diets with the advancement of effective in vitro synergistic synbiotic identification methods [335]. Additionally, it would be beneficial for studies to state the rationale behind combinations utilized and to use the appropriate synbiotic subset name in publications. Future research is warranted to firstly identify additional effective synbiotic combinations, perhaps involving the use of novel substrates, and secondly to verify their effectiveness in vivo in comparison to probiotic supplementation alone. A summary of recent publications in the literature detailing the effects of maternal sow synbiotic supplementation on sow and offspring microbiota and health and performance parameters are presented in Table 3.

## 4. Conclusions

The increasing concern surrounding the potential development of antimicrobial-resistant pathogens has prompted the goal of reducing antibiotics and antimicrobials in commercial pig production. Recent research has shown that including probiotic or probiotic blends in pigs’ diets has multiple beneficial effects via direct and/or indirect mechanisms. The importance and plasticity of the pioneer microbiota, in combination with the dominant role of the maternal microbiota in its establishment, indicate modulating the sow’s microbiota as a critical area of exploration. Maternal probiotic or synbiotic supplementation offers an effective strategy to modulate the sow’s microbiota and thereby enhance the formation of a health promoting pioneer microbiota in the offspring. In addition, this strategy can also reduce oxidative stress and inflammation in the sow and her offspring, enhance the immune potential of the milk, the immune system development in the offspring and the sows feed intake during lactation. In theory, synergistic synbiotic supplementation has the greatest potential and the widest scope, with the combination of a beneficial microorganism and a substrate amplifying the potential benefits compared to either alone. To date, investigations analysing maternal synbiotic supplementation are sparce, and has only included complimentary synbiotic combinations, while synergistic synbiotic or synbiotic combinations with newly classed prebiotics have yet to be explored. Further research investigating maternal probiotic or synbiotic supplementation is warranted, particularly on the effects on lifetime offspring performance and the role early gut colonizers play in the longterm shaping of the gut microbiota. In addition, there are several relatively novel parameters, such as AhR ligand production and the microbiota, oligosaccharide, and microbial metabolite content of the sow’s milk, highlighted in this review, that may be influenced by maternal probiotic or synbiotic supplementation that demands attention in future investigations.

## Figures and Tables

**Figure 1 animals-13-02996-f001:**
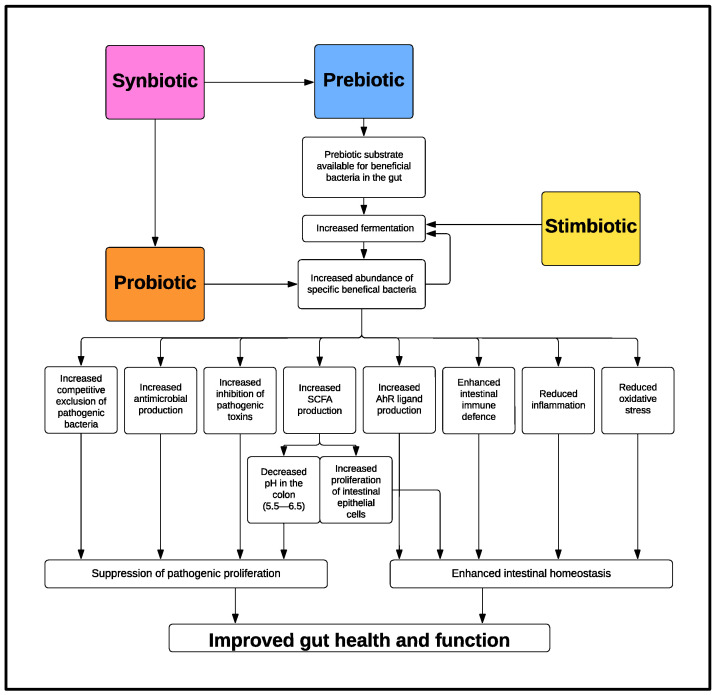
Mode of action of probiotics, prebiotics, synbiotics, and stimbiotics in the GIT.

**Table 1 animals-13-02996-t001:** Microorganisms and their characteristics commonly used as probiotics in pig diets.

Genus	Species	Bacterial	Spore Forming	Autochthonous (Previously Isolated from Pig)	References *
*Bacillus*	*B. altitudinis*	✓	✓	X	[2]
	*B. amyloliquefaciens*	✓	✓	X	[52]
	*B. cereus*	✓	✓	X	[53]
	*B. licheniformis*	✓	✓	X	[54]
	*B. subtilis*	✓	✓	X	[55]
*Bifidobacterium*	*B. animalis*	✓	X	✓	[56]
	*B. longum*	✓	X	X	[57]
	*B. thermophilum*	✓	X	✓	[58]
*Clostridium*	*C. butyricum*	✓	✓	✓	[59]
*Escherichia*	*E. coli Nissle*	✓	X	✓	[60]
*Enterococcus*	*E. faecium*	✓	X	✓	[61]
	*E. faecalis*	✓	X	✓	[61]
*Lactobacillus*	*L. acidophilus*	✓	X	✓	[62]
	*L. amylovorus*	✓	X	✓	[63]
	*L. casei*	✓	X	✓	[64]
	*L. johnsonii*	✓	X	✓	[64]
	*L. plantarum*	✓	X	✓	[62]
	*L. reuteri*	✓	X	✓	[64]
	*L. rhamnosus*	✓	X	X	[65]
	*L. salivarius*	✓	X	✓	[66]
*Lactococcus*	*L. lactis*	✓	X	✓	[67]
*Pediococcus*	*P. acidilactici*	✓	X	✓	[28]
	*P. pentosaceus*	✓	X	✓	[64]
*Saccharomyces*	*S. cerevisiae*	X	✓	X	[68]

* A tick (✓) in column 5 refer to studies where the microorganism has been isolated from pig. In cases where there was no available reference for the microorganism isolation from pig (X in column 5), a reference to a study where the microorganism was utilised as a probiotic in pigs is included.

**Table 2 animals-13-02996-t002:** Effects of probiotic inclusion in the maternal diet on sow and offspring GIT microbiota and health/performance parameters.

Probiotic Strains	Main Sow Effects	Main Offspring Effects	Reference
*Bacillus altitudinis*	*Bacillus altitudinis* was not detected in sow faeces prior to probiotic administration. After maternal supplementation it was detected in sow faeces and increased from D100 gestation to D13 of lactation. No impact on sow faeces alpha diversity or differences in abundance at phylum, family, or genus level. Increased total solids and protein and reduced lactose in colostrum. Reduction in bacterial diversity in the colostrum, increased relative abundance of *Actinobacteriota* and *Rothia*.	*Bacillus altitudinis* detected in the offspring faeces when tested on D13 of lactation. Reduced abundance of *Synergistota*, *Alloprevotella and Rikenellacea dgA-11gut group* in faeces at D26 of lactation. Increased the relative abundance of *Cantenibacterium*, *Blautia*, *Rikenellaceae* RC9 gut group, *Prevotella* and *Prevotella* NK3B31 group in ileum on D8 PW. Increased *Lactobacillus* in faeces on D118 PW. Increased abundance of bacterial taxa associated with breakdown of complex carbohydrates and nutrient utilization in the ileum at D8 PW. Tendency for increased butyric acid in ileum digesta on D8 PW. Improved offspring FCR D0–D14 PW and increased bodyweight at D105 and D127 PW. Improved carcass weight and kill-out percentage.	[2,20]
*Bacillus subtilis*	Effect on sow microbiota not recorded. Increased total born and born alive.	Increased *Lactobacillus gasseri* and *Lactobacillus johnsonii* in ileum and reduced *Escherichia coli* in colon of offspring at D3 of lactation. Increased *Lactobacillus* in colon of offspring at D10 of lactation. Increased offspring weaning weights and tendency to improve ADG. Tended to increase number of piglets weaned.	[13]
*Bacillus subtilis*	Reduced alpha diversity in sow faeces at D110 of gestation. Increased the relative abundance of phyla *Gemmatimonadete* and *Acidobacteria* and reduced relative abundances of *Actinobacteria*, *Proteobacteria*, and *Streptococcus*.	Effect on offspring microbiota not recorded. Increased total born and number born alive but decreased birth weights. Increased litter weight and number of pigs per sow at weaning.	[296]
*Bacillus subtilis* C-3102	Increased total number of *Bacillus* sp. in sows’ faeces.	Increased total number of *Bacillus* sp. in piglets’ faeces. Reduced ADG, ADFI and BW of offspring in late nursery.	[14]
*Bacillus subtilis* C-3102	Positive for probiotic strain in faeces at all stages. Reduced *Escherichia coli* in faeces on D110 of gestation in second cycle. Increased sow feed intake, increased sow BCS during pregnancy, reduced sow weight loss during lactation, reduced WSI.	Tendency for reduced *Clostridium* spp. in the offspring faeces in the first cycle. Reduced *Escherichia coli* and *Clostridium* spp. in offspring faeces in second cycle. In second cycle of supplementation there was an increase in birth weight and number of pigs weaned.	[292]
*Bacillus subtilis* or *Bacillus amyloliquefaciens*	Both probiotics reduced alpha diversity in sow faeces on D8 of lactation. Both probiotics reduced the abundance of *Prevotellaceae*, *Lachnospiraceae*, *Ruminococcaceae*, and *Bacteroidaceae* families in sows’ faeces on D8 of lactation. Both probiotics reduced *Bacteroides*, Faecalibacterium, *Phascolarctobacterium*, *Prevotella*, *Blautia*, *Dorea*, and *Roseburia*.	*Bacillus subtilis* reduced alpha diversity on D21 of lactation compared to control and *Bacillus amyloliquefaciens*. The majority of effects of maternal probiotic treatment were observed at minor taxa in the offspring faeces and the effect was greater PW than preweaning. Postweaning both probiotic groups had greater abundances of *Ruminococcaceae* and reduced p-2534-18B5. *Bacillus amyloliquefaciens* increased the abundance of *Bacteroidales* BS11 gut group and F082. A high number of positive correlations were observed between the maternal microbiota and the microbiota of the weaned offspring. *Bacillus amyloliquefaciens* increased number of pigs born alive and weaned.	[52]
*Bacillus subtilis* and *Bacillus amyloliquefaciens* (supplemented to both sows and offspring)	Effect on microbiota not recorded. Increased IgG in sows’ blood at farrowing.	Effect on microbiota not recorded. Improved offspring weaning weight, increased total creep feed consumption and litter gain. Increased IgM in blood at weaning. Increased villus height and Peyer’s patch size.	[297]
*Bacillus licheniformis* and *Bacillus subtilis* (BioPlus 2B^®^)	Effect on microbiota not recorded. Decreased number of repeat sows after service in experimental group. Increase in sow feed intake D1–D14 of lactation. Reduced sow weight loss. Increased fat and protein content of milk.	Effect on microbiota not recorded. Decreased offspring diarrhoea score and mortality and increased number of pigs per sow and weight at weaning. Increased creep feed intake in offspring.	[294]
*Bacillus licheniformis* and *Bacillus subtilis* (BioPlus 2B^®^)	Effect on microbiota not recorded.	Effect on microbiota not recorded. Reduced diarrhoea and increased BW at weaning.	[54]
*Bacillus licheniformis* and *Bacillus subtilis* (BioPlus 2B^®^)	Effect on microbiota not recorded. Reduced sow weight loss. Increased sow feed intake.	Effect on microbiota not recorded. Increased litter weaning weight.	[293]
*Bacillus licheniformis* and *Bacillus subtilis* (0.1 and 0.2% inclusion rate)	Inclusion rate of 0.2% reduced faecal *Escherichia coli* population at weaning. Inclusion rate of 0.2% improved DM digestibility and decreased faecal NH_3_ concentration.	Effect on offspring microbiota not recorded. Linearly increased BW at D21 and D25 of lactation and ADG throughout lactation compared to control.	[295]
*Bacillus licheniformis* and *Bacillus subtilis* A or B	No change in sow faeces microbiota quantities. Increased colostrum fat content.	Effect on offspring microbiota not recorded. Increased average weaning weight and ADG.	[55]
*Bacillus subtilis* A and *Bacillus subtilis* B	Increased *Firmicutes* on D7 of lactation in sow faeces. Increased total SCFA on D7 of lactation in sow faeces. Increased colostrum fat content.	Effect on offspring microbiota not recorded. Increased average weaning weight and ADG.	[55]
*Bacillus subtilis* and *Lactobacillus acidophilus*	Effect on microbiota not recorded. Increased sow feed intake during lactation and sow backfat thickness at weaning.	Effect on microbiota not recorded. Increased piglet birthweight.	[291]
*Clostridium butyricum* (0.1, 0.2 and 0.4% inclusion rate)	0.2% probiotic increased relative abundance of *Bacteroidetes* and decreased relative abundance of *Proteobacteria*, *Gemmatimonadetes* and *Actinobacteria* at phylum level. At genus level there were notable changes, particularly in the increased abundance in Prevotella in sow faeces. Increasing concentration of probiotic: linearly decreased interval between piglet born and duration of farrowing quadratically. Linearly increased colostrum IgG and IgM. Decreased sow serum MDA concentrations at parturition and D14 of lactation.	Effect on offspring microbiota not recorded. Increasing concentration of probiotic linearly increased in litter weight gain and litter weaning weight. Decreased piglet serum MDA concentrations at D14 and D21 of lactation.	[29]
*Clostridium butyricum*, *Bacillusmesentericus,* and *Enterococcus faecalis* in a peptide-zinc compound (BIO-THREE PZ) (Supplemented to sows infected with PEDV)	Effect on microbiota not recorded. Increased sow bodyweight at D7 of lactation. Increased milk production and IgA concentration in milk. Reduced WSI.	Effect on microbiota not recorded. Increased piglet birthweight.	[260]
*Enterococcus faecium*	Minor changes with slight reduction in *Escherichia coli* and slight increase in *Lactobacillus* counts in sow faeces. Increased sow apparent total digestibility of dry matter, nitrogen, and gross energy.	Increasing probiotic concentration linearly increased *Lactobacillus* and *Enterococci* and linearly decreased *Escherichia coli* in offspring faeces on D14 PW. Increasing probiotic concentration linearly reduced offspring diarrhoea score and preweaning mortality and linearly increased offspring ADG, G: F and weaning weight.	[48]
*Enterococcus faecium* vs. *Bacilluscereus*	Effect on microbiota not recorded. *Bacilluscereus* increased sow faecal IgA before weaning.	Effect on microbiota not recorded. *Enterococcus faecium* reduced offspring faecal IgA D7 PW. Both probiotics decreased serum IgG postweaning. *Bacilluscereus* increased offspring faecal IgA before weaning.	[298]
*Lactobacillus casei*	Effect on microbiota not recorded.	Effect on microbiota not recorded. Increased litter weight at weaning, average weaning weight, and piglet weight gain.	[299]
*Lactobacillus johnsonii* XS4	Viable count of faecal flora showed no difference in sows. Increased sow serum IgG.	Effect on offspring microbiota not recorded. Increase in litter birth weight, litter weight at D20 of lactation and weaning litter weight.	[300]
*Lactobacillus plantarum*	Effect on microbiota not recorded. No effect on reproductive performance of sow.	Effect on microbiota not recorded. Reduced preweaning mortality and diarrhoea and increased weight gain in offspring.	[43]
*Lactobacillus reuteri*	Increased abundance of *Anaerostipes* at D101 of gestation, *Lachnospiracea_incertae_sedis* and *Mogibacteriumat* at D115 of gestation, Ruminococcus at D7 of lactation, and Acinetobacter at D14 of lactation. Altered microbiota of the colostrum: decreased the abundance of the phylum *Proteobacteria* and increased that of the species *Bifidobacterium choerinum*. Altered the metabolites present in the umbilical cord blood serum.	Increased alpha diversity and abundance of 21 bacterial taxa at genus level in the meconium of offspring. Increased SOD and decreased MDA level in piglet serum on day 21 of lactation.	[198]
*Lactobacillus plantarum* and *Saccharomyces cerevisiae*	Increased sow faecal abundance *Ruminococcus*, *Bacteroides*, and *Anaeroplasma* and decreased *Tenericutes* on day 105 of gestation. Increased the abundances of *Actinobacteria* and *Anaerostipes* and decreased *Proteobacteria* and *Desulfovibrio* on D21 of lactation. Decreased the faecal levels of tryptamine, putrescine, and cadaverine on D105 of gestation and isovalerate and skatole on D21 of lactation while increased butyrate level on D21 of lactation.	Increased abundance of *Deferribacteres*, *Fusobacteria*, and *Fusobacterium* and decreased *Anaerostipes* in the offspring’s colon. Reduced oxidative stress and inflammation response markers in plasma.	[223]
*Lactobacillus plantarum* and *Saccharomyces cerevisiae*	Effect on sow microbiota not recorded.	At D65 of age there was a remarkable increase in relative abundance of *Gemmiger*, while the relative abundance of *Roseburia* and *Blautia* tended to increase. Probiotic supplementation altered the metabolism pathway of carbohydrate, amino acids, cofactors, and vitamins in the colonic microbiota.	[301]
*Pediococcus acidilactici*	Increased *Lactobacilli* and reduced *Escherichia coli* and *Staphylococcus aureus* populations in sow faeces at weaning. Increased serum concentrations of IgG, IgA, and total protein. Decreased serum haptoglobin concentration and alanine aminotransferase activity at weaning	Effect on offspring microbiota not recorded. Increased the number of piglets weaned, increased average piglet weight, litter weight and survival rate at weaning. Decreased diarrhoeal rate of piglets during lactation.	[302]
*Saccharomyces cerevisiae*	No change in faecal microbial alpha or beta- diversity of sows. Increased abundance *Ruminococcaceae_UCG-005*, *Ruminococcus_1*, *Prevotellaceae_UCG001*, *Family_XIII_AD3011*, *Acetitomaculum,* and *Lachnospiraceae_NK4B4.* Increased sow feed intake. Reduced WSI. Increased plasma ghrelin and IgG and decreases glucagon like peptide-1 at D110 of gestation. Increased milk production in first week of lactation.	Effect on offspring microbiota not recorded. Increased offspring ADG.	[290]
*Saccharomyces cerevisiae*	Effect on microbiota not recorded. Increased protein, lactose, and solids-not-fat in colostrum. Increased plasma IgG in sows on D1 of lactation.	Effect on microbiota not recorded. Reduced stillborn and low birth weight piglets.	[303]
*Saccharomyces cerevisiae* (Actisaf Sc 47^®^)	Effect on microbiota not recorded. Reduced WSI. IgG tended to increase in colostrum.	Effect on microbiota not recorded. Increased IgG on D1 of lactation.	[261]
*Saccharomyces cerevisiae* (Actisaf Sc 47^®^)	Effect on microbiota not recorded. Reduced uterus and/or udder disease in sows.	Effect on microbiota not recorded. Reduced incidence of diarrhoea in offspring. Increased litter size and litter weight at weaning.	[304]
*Saccharomyces cerevisiae*	Effect on microbiota not recorded. Increased sow serum concentration of IgA and IgG at farrowing and weaning.	Effect on microbiota not recorded. Increased offspring serum concentration of IgA and IgG on D14 and D28 PW.	[68]
*Saccharomyces cerevisiae*	Effect on microbiota not recorded. Increased daily feed intake, total feed intake, milk production, and fatty acid profile of the milk. Tendency to reduce weight loss.	Effect on microbiota not recorded. Tended to increase number of born alive. Increased weaning weight and average daily gain.	[289]
*Saccharomyces cerevisiae*, *Lactobacillus casei*, *Lactobacillus plantarum*, *Enterococcus faecium*, *Enterococcus faecalis*, *Bifidobacterium bifidum*, *Bifidobacterium pseudolongum*, *Bacillus licheniformis*, *Bacilluscereus var toyoi*, *Bacillus subtilis*, *Clostridium butyricum* (Bokashi^®^)	Effect on microbiota not recorded. Increased IL-2 in sow serum on D114 of gestation. Increased IL-6 in sow serum on D60 of gestation. Increased IL-4 and IL-10 in sow serum. Increased TGF-β, IgG and IgA in sow serum on D114 of gestation. IL-2 and IFN-γ were greater in the colostrum and milk at all sampling times. Increase in sow bodyweight at farrowing. Reduced sow weight loss during lactation.	Effect on microbiota not recorded. Increased number of born alive. Reduced percentage of offspring with diarrhoea and reduced mortality rate in the first 7 days of lactation.	[235]

ADFI: average daily feed intake, ADG: average daily gain, BCS: body condition score, BW: bodyweight, DM: dry matter, D0: Day 0, FCR: feed conversion ratio, G:F: gain to feed, Ig: Immunoglobulin, IFN: interferon, IL: interleukin, MDA: malondialdehyde, PW: postweaning, SCFA: short chain fatty acid, SOD: superoxide dismutase’s, TGF: transforming growth factor, WSI: wean to service interval.

**Table 3 animals-13-02996-t003:** Effects of synbiotic inclusion in the maternal diet on sow and offspring GIT microbiota and health and performance parameters.

Synbiotic	Main Effects on Sow	Main Effects on Offspring	Reference
Fructo-oligosaccharide, *Enterococcus faecium*, *Pediococcus acidilactici*, *Bifidobacterium animalis*, and *Lactobacillus reuteri*	Effect on microbiota not recorded.	Effect on microbiota not recorded. Increased litter weight at weaning. Trend for improved number of pigs weaned per sow and litter weight gain.	[336]
Xylo-oligosaccharide, *Lactobacillus plantarum* and *Saccharomyces cerevisiae*	Increased faecal Simpson and Shannon indices, and the relative abundances of *Coprococcus*, *Clostridium*, *Ruminococcus,* and *Prevotella* on D105 of gestation. Increased relative abundance of *Proteobacteria* on D21 of lactation but decreased *Desulfovibrio* and *Herbaspirillum*. Slightly decreased faecal SCFA. Decreased colostrum somatic cell numbers. Decreased fat and total dry milk matter. Altered lipid metabolism indicators.	No change in alpha diversity of colonic content microbiota. Increased abundances of Sphingomonas, *Anaerovorax*, *Butyricicoccus,* and *Sharpea* in colonic content. Decreased propionate level in the colonic content. Increased piglet survival rate. Increased plasma isoleucine and leucine levels and decreased taurine, cysteine, and alanine. Reduced oxidative stress. Decreased pro-inflammatory cytokine levels in the plasma. Decreased ghrelin, cholecystokinin, and pancreatic polypeptide levels in plasma. Upregulated mRNA tight-junction proteins.	[337,338]
Xylo-oligosaccharide, *Lactobacillus plantarum* and *Saccharomyces cerevisiae*	Did not record effects on the sow.	In offspring at 65 days of age: increased relative abundance of *Actinobacteria*, *Bifidobacterium*, *Turicibacter*, and *Clostridium* in the jejunum compared with an antibiotic treated group. In offspring at 65 days of age: increased serum IgA, jejunal IL-10, interferon- α, and sIgA concentrations. Increased ileal villus height.	[339]
Xylo-oligosaccharide, *Lactobacillus plantarum* and *Saccharomyces cerevisiae*	Did not record effects on the sow.	In offspring on day 30 postweaning: increased abundance of *Firmicutes*, *Bacteroidetes*, *Lactobacillus* and *Bifidobacterium* in the jejunum. In offspring on day 30 PW: improved antioxidant activity in serum and in jejunal mucosa.	[15]
Xylo-oligosaccharide, *Lactobacillus plantarum* and *Saccharomyces cerevisiae*	Did not record effects on the sow.	No effect on alpha diversity indices of colonic content in the offspring at D65, 95, or 125 of age. Beta-diversity of colonic content showed distinction at D65 of age. Decreased colonic SCFA concentration at D95 of age. Colonic concentration of indole increased and skatole decreased at D65 of age. Affected metabolism of carbohydrate, amino acid, cofactors, and vitamins in the colonic microbiota.	[301]

D105: day 105, mRNA: messenger ribonucleic acid, SCFA: short chain fatty acids.

## Data Availability

Not applicable.

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
