# Peer review of "The Effect of Maternal Probiotic or Synbiotic Supplementation on Sow and Offspring Gastrointestinal Microbiota, Health, and Performance"

_animals, 2023, doi:10.3390/ani13192996_

Round 1
Reviewer 1 Report
The manuscript provided useful information regarding gut health, antimicrobial stewardship, and pig production. This review is important and the readers will find interesting information about sustainable pig production. Despite its potential impact, some minor issues need to be addressed.
Title
Line 2-3: ‘Microbiome’ should be replaced with ‘gut microbiome’. This is necessary because the manuscript centers only on the gut microbiome and not other microbiomes viz: URT, reproductive, oral, nasal, etc.
Abstract
The abstract should be improved. The punctuations are a bit poor. The authors should also summarize the major findings in the abstract and perhaps a conclusion and/or future direction.
Content
Authors should briefly discuss the impact of gut health (and its modulation) in poultry production, especially in weight gain, egg production, FCR, haemato-biochemical parameters, etc.
The authors should:
1. Discuss the overall gut anatomy and gut physicochemical properties of the microbiome of a healthy pig.
2. Authors should be able to differentiate between ‘microbiome’ and ‘microbiota’ and then concentrate on which they intend to address. If the authors insist on discussing the microbiome, then they have to also discuss other relevant components of the gut microbiome especially, nucleic acids, peptides/proteins, mobile genetic elements, metabolites (short chain fatty acid, etc.), organic acids, etc.
3. Mechanisms of symbiotic action and their impacts on health and performance could be well discussed, too
Minor edits, especially on the abstract are required.
Author Response
Dear Reviewer,
Thank you for reviewing our manuscript.
Please see the attached document for responses to suggestions.
Best wishes,
Dillon Kiernan

Reviewer 2 Report
The aim of the paper to review current literature related to maternal probiotic or synbiotic supplementation on sow and offspring was achieved. The writing in the paper is generally clear and provides a good review of the topic. There are a few changes suggested to improve the paper below. However, the review covers the topic quite well and clearly indicates the relevance of the review topic to the swine industry.
Specific comments:
Line 19 - replace "serval" with "several"
Line 21 - replace "serval" with "several"
Line 30 - remove comma after "microbiome"
Line 49 - replace "therapeutic" with "pharmacological"
Line 53 - remove "early"
Line 62 - remove "prematurely"
Line 63 - insert "three to" before "four"
Line 64 - replace "Hence" with "In addition"
Line 79 - remove comma after "microbiome"
Line 118 - replace "though" with "through"
Line 194 - replace "can't" with "cannot"
Line 336 - remove ", they are" before "produced"
Line 355 - remove "s, which are" before "glycoproteins"
Line 336 - replace "which that are" with "as"
Line 374 - replace "While" with "For example"
Line 376 - replace "Contrary to this, pigs" with "Pigs"
Line 377 - replace ", while" with ". However"
Line 379 - replace "While, pre" with "Pre"
Lines 381-383 - reword - this sentence does not make sense
Line 386 - remove comma after "proteins"
Line 386 - remove "cells" after "immune"
Line 387 - remove "also" before "intestinal"
Line 390 - remove "established" before "negative"
Line 397 - replace "While, pre" with "Pre"
Line 409 - replace "While, environmental" with "Environmental"
Line 437 - replace "and ," with comma
Table 2 - page 19 - main offspring effects: explain "dgA-11gut group"
Table 2 - page 20 - main offspring effects: replace "affect" with "effect"

Author Response

(The authors gave the same response as above.)

Reviewer 3 Report
Dear Editor,
It is my pleasure to review the manuscript written by Dr Kiernan et al entitled “The effect of maternal probiotic or synbiotic supplementation on sow and offspring microbiome, health, and performance.”. In the review paper, authors discussed a hot and important topic that is related to prevalence of antimicrobial-resistant pathogens and reduction of antibiotic and antimicrobial use in commercial pig production. Authors discussed possibilities that how to increase animal health and performance through promoting an improved GIT microbiome, particularly the pioneer microbiome in the young pig, is a fundamental focus. It also discussed about dietary manipulation of the sow’s microbiome with probiotics or synbiotics, prior to farrowing and during lactation, is a compelling area of exploration. Supplementing with a synbiotic and probiotic, and prebiotic substrated was also discussed regarding their benefits to the health of the sow's microbiome.
In general, the topic is very interesting and very important, and the information in the review paper is quite comprehensive. I have several comments to authors before the manuscript is published.
Question 1: Line 49-51, whether EU release the latest version regarding regulation? In addition to EU regulation, authors may also discuss regulations from other important regions.
Question 2: Please add citations for the sentence from line 62-64;
Question 3: For table 1, authors described probiotics used pig diets, whether different probiotics fit for different stages of pig growth? E.g. piglets, nursing, sow, lactating and so on.
Question 4: these is a mistake for numbersing “Probiotics”
Question 5: if possible, I suggest authors make a comparison table or figure to interpret similarity and differences of probiotic and synbiotic for GIT microbiome.
Moderate revision
Author Response

(The authors gave the same response as above.)

Round 2
Reviewer 3 Report
thank you so much for revising the study, i do not have further questions. thanks